# Acceleration-induced spectral beats in strongly driven harmonic oscillators

A. S. Kuznetsov [1], K. Biermann[1] & P. V. Santos [1] ✉

The harmonic modulation of coherent systems gives rise to a wealth of physical phenomena, e.g., the AC-Stark effect and Mollow triplets, with important implications for coherent control and frequency conversion. Here, we demonstrate a novel regime of temporal coherence in oscillators harmonically driven at extreme energy modulation amplitudes relative to the modulation quantum. The studies were carried out by modulating a confined exciton-polariton Bose-Einstein condensate (BEC) by an acoustic wave. Features of the new regime are the appearance, in the spectral domain, of a comb of resonances termed *acceleration beats* with energy spacing tunable by the modulation amplitude and, in the time domain, of temporal correlations at time scales much shorter than the acoustic period, which also depend on the modulation amplitude. These features are quantitatively accounted for by a theoretical framework, which associates the *beats* with accelerated energy-change rates during the harmonic cycle. These observations are underpinned by the high sensitivity of the BEC energy to the acoustic driving, which simultaneously preserves the BEC's temporal coherence. The *acceleration beats* are a general feature associated with accelerated energy changes: analogous features are thus also expected to appear under highly accelerated motion e.g., in connection with Cherenkov and Hawking radiation.

How does a state with the finite temporal coherence respond to a harmonic modulation with angular frequency $\Omega_M$? This fundamental question appears in a wide range of scenarios including the rich physics connected to the AC-Stark effect[1] as well as to the spectral emission of fast moving objects such as electrons in solids and matter attracted by black holes. A simple answer is based on the ratio between the state ($s$) energy decoherence rate, $\gamma^{(s)}$, which determines the spectral linewidth, and the modulation quantum $\hbar\Omega_M$. When $\gamma^{(s)} > \hbar\Omega_M$, the state adiabatically follows the imposed energy modulation: its time-integrated spectral response then turns into a continuous "camelback" spectrum with maxima separated by the peak-to-peak energy modulation amplitude. On the contrary, if the condition $\gamma^{(s)} < \hbar\Omega_M$ is fulfilled by either increasing $\Omega_M$ or reducing $\gamma^{(s)}$, one enters the non-adiabatic regime[1], where the state responds by emitting and absorbing an integer number of $\hbar\Omega_M$ quanta. The main spectral signature is then the emergence of a comb of sideband resonances shifted from the original one by multiples of $n \times \hbar\Omega_M$ with $n = 0, \pm 1$, and so on. The modulation amplitude can be expressed as $\chi^{(s)}\hbar\Omega_M$, where the dimensionless energy modulation index $\chi^{(s)}$ determines the sideband amplitudes according to $J_n(\chi^{(s)})$, where $J_n$ is the Bessel function of the first kind and order $n$. The non-adiabatic regime is particularly relevant for optomechanical applications, as it enables the coherent conversion between microwave and optical photons, the generation of frequency combs as well as optical heating, cooling, amplification and sensing of mechanical motion[2]. In this context, it is most commonly understood that reaching the non-adiabatic regime necessarily requires long-living states (small $\gamma^{(s)}$) and/or high-frequency modulation (large $\Omega_M$).

Here, we experimentally demonstrate that harmonically driving the energy $E_M(t)$ of a resonator at high amplitudes (typically, at

[1]Paul-Drude-Institut für Festkörperelektronik, Leibniz-Institut im Forschungsverbund Berlin e. V., Hausvogteiplatz 5-7, 10117 Berlin, Germany.
✉e-mail: santos@pdi-berlin.de

$\chi^{(s)} > 100$) leads to a novel regime of non-adiabaticity, which, counter-intuitively, is attained even when $\gamma^{(s)} > \hbar\Omega_M$. Specifically, even if non-adiabatic features are absent for weak driving, they can emerge and dominate the spectral response for large indices $\chi^{(s)}$. We show that this regime emerges from non-adiabatic effects induced by high accelerated rates of energy change $|\partial^n E_M(t)/\partial t^n|$ ($n = 2, \dots$), which induce temporal correlations at a time scale much shorter than the modulation period. Their main spectral signature is the emergence of *acceleration beats* – spectral oscillations related to these temporal correlations with the energy spacing determined by $\chi^{(s)}$. Contrary to the conventional sidebands observed for small energy modulation indices $\chi^{(s)}$, which arise from energy change rates linear in time and have a fixed energy separation $\hbar\Omega_M$, the *acceleration beats* arise from the bunching of sidebands to form combs with an energy spacing that increases with $\chi^{(s)}$. This spacing can exceed the state decoherence, thus leading to a non-adiabatic response even for $\hbar\Omega_M < \gamma^{(s)}$.

Interestingly, the *acceleration beats* are predicted already by the simple Bessel function model for the harmonic modulation of a coherent state. The beats appear as an envelope for the sidebands at the extrema of the energy modulation cycle for large $\chi^{(s)}$'s. Non-adiabatic modulation schemes yielding a large number of sidebands (i.e., with $\chi^{(s)} > 100$) have been reported for a few physical systems[3,4]. However, to the best of our knowledge, these beats have so far not been reported. Here, one of the reasons is that their observation requires large energy modulation amplitudes relative both to the modulation quantum and the decoherence rate, which must be achieved while maintaining the temporal coherence and the harmonic character.

A convenient approach to induce high harmonic modulation amplitudes exploits the strain-induced modulation of spectrally narrow optical resonances by piezoelectrically excited high-frequency acoustic waves. These waves have been applied to generate non-adiabatic sidebands in long-living atomic-like states such as those in single color centers[5,6] and quantum dots[7–11]. Non-adiabatic modulation by fast strain fields has also been reported for microcavity polaritons[12] as well as for the confined microcavity polariton condensates investigated here[13,14].

Polaritons are light–matter particles resulting from the strong coupling between photons and quantum well excitons in a semiconductor microcavity (MC). At low particle densities, the polariton coherence is determined by the exciton decoherence and the cavity photon lifetime, leading to spectral linewidths of typically a few hundreds of $\mu$eV. At high densities, polaritons undergo a transition to a highly coherent *non-equilibrium* Bose–Einstein condensate (BEC) with linewidths down to a few $\mu$eV and, thus, temporal coherences reaching the ns-range. In the BEC regime, polaritons retain an enhanced coupling to strain fields via their excitonic component, which enables efficient energy modulation by acoustic waves. Here, we use highly coherent confined BEC states (i.e., with coherence time $\sim \hbar/\gamma^{(s)} \approx 1.7$ ns[13,14]) modulated by sub-GHz surface acoustic waves (SAWs) to demonstrate *acceleration beats* in an harmonically driven optomechanical system. We show that SAWs can induce very high energy modulation indices $\chi^{(s)} > 200$ without introducing decoherence. These high amplitudes induce additional spectral and temporal features in the BEC response, which are attributed to the *acceleration beats*. This conclusion is fully supported by a theoretical framework, which proves that the beats arise from the accelerated energy change rates. The combination of strong interaction with vibrations together with the optical detection and acousto-electrical control makes confined polariton BECs excellent candidates for the simulation of acceleration-induced coherent effects.

## Results

### Acoustic modulation of confined polaritons
The studies were carried out on a nominally $4 \times 4$ $\mu$m$^2$ potential trap for polariton BECs fabricated within the spacer region of an (Al,Ga)As MC. The trap consists of a thicker mesa region lithographically defined in the MC spacer [cf. Fig. 1a]. The latter reduces the photon energies, thus providing the lateral confinement potential for polaritons[15]. The acoustic modulation experiments were carried out at 10 K on a polariton trap placed in a SAW resonator consisting of two interdigital acoustic transducers [IDTs, cf. Fig. 1b]. One of the IDTs is driven by an radio-frequency (rf) $f_M = \Omega_M/(2\pi) = 384$ MHz with tunable power $P_{rf}$ while the other IDT acts as a passive acoustic reflector. The rf signal excites a SAW with amplitude $A_M \propto \sqrt{P_{rf}}$. The trap was placed close to an anti-node of the standing hydrostatic strain field in order to maximize the deformation-potential interaction between excitons in the trap and the SAW strain field[16]. Details of the MC fabrication process and acoustic excitation can be found in Methods.

The lateral confinement creates the discrete polariton states illustrated in the spatially resolved photoluminescence (PL) map of Fig. 1c. This map was acquired under non-resonant continuous wave laser excitation with a low optical excitation power, $P_{exc}$ (see Methods). The dashed lines in Fig. 1c depict the trap confinement potential and corresponding wave functions calculated for the lower confined levels. The latter were determined by taking into account the trap dimensions obtained from atomic force microscopy profiles of the final MC surface, which retain the mesa shapes defined in the spacer [cf. inset of Fig. 1b][17]. The spectrum of the confined states agrees very well with those expected for polaritons in the confinement potential[17].

The evolution of the confined ground (GS) and first excited (ES) states with increasing $P_{exc}$ is depicted in Fig. 1d: here, the confined levels initially blueshift with increasing $P_{exc}$ until polariton BEC sets in at the threshold power density $P_{th} = 24$ kW/cm$^2$. A spatial PL map in the BEC state is displayed in Fig. 1e. Condensation occurs both in the GS and first excited state (ES): the emission intensity of these states increases by orders of magnitude at condensation while their linewidths decrease to the GHz range, as will be discussed below. The simultaneous formation of condensates both in the GS and ES is a consequence of the non-equilibrium nature of polariton BECs. Unlike a conventional BEC in thermodynamical equilibrium, polariton BECs result from the dynamic balance between particle losses (e.g., due to the photon escape from the MC) and replenishment by stimulated scattering from states excited by the pumping laser. This non-equilibrium process enables the formation and coexistence of BECs in different confined states of the trap with particle densities dictated by the dynamics of the stimulated scattering and particle loss mechanisms.

The impact of the acoustic field on the BEC states is illustrated by the time-resolved PL map of Fig. 1f, which was recorded using a streak-camera. Here, the trap was subjected to a standing SAW with a nominal amplitude $A_M = 0.56 \sqrt{W}$ generated by rf excitation of one of the IDTs. The energy of the confined states oscillates as a function of time with the SAW periodicity $T_M = 1/f_M \approx 2.6$ ns with maximum shifts of $\pm 0.5$ meV. In agreement with previous studies[16], the energy modulation amplitude is mainly determined by the strain-induced changes of the excitonic energies via the deformation potential (DP) mechanism (i.e., the changes of the excitonic energies induced by a strain field, see, for details, Supplementary Note 2).

### High-resolution spectroscopy
In the absence of acoustic excitation, the BECs in the GS and ES have a single spectral line (the zero-phonon-line) with linewidth (defined as the full width at half maximum, FWHM) $\gamma^{(s)}/\hbar = 0.6$ GHz, as was determined from the high-resolution spectrum recorded using a tunable etalon displayed in Fig. 2 (see Methods and Supplementary Note 1 for details about the measurement procedure). These linewidths are significantly narrower than the ones of approx. 50 GHz measured for excitation fluences below the BEC threshold [cf. Fig. 1c]. The remaining time-integrated PL spectra in Fig. 2 were recorded under different acoustic amplitudes $A_M$. With increasing $A_M$, the oscillator strength of the emission is progressively transferred from the zero-phonon line towards the extrema $\pm(\chi^{(GS)}\hbar\Omega_M)$ of the energy modulation

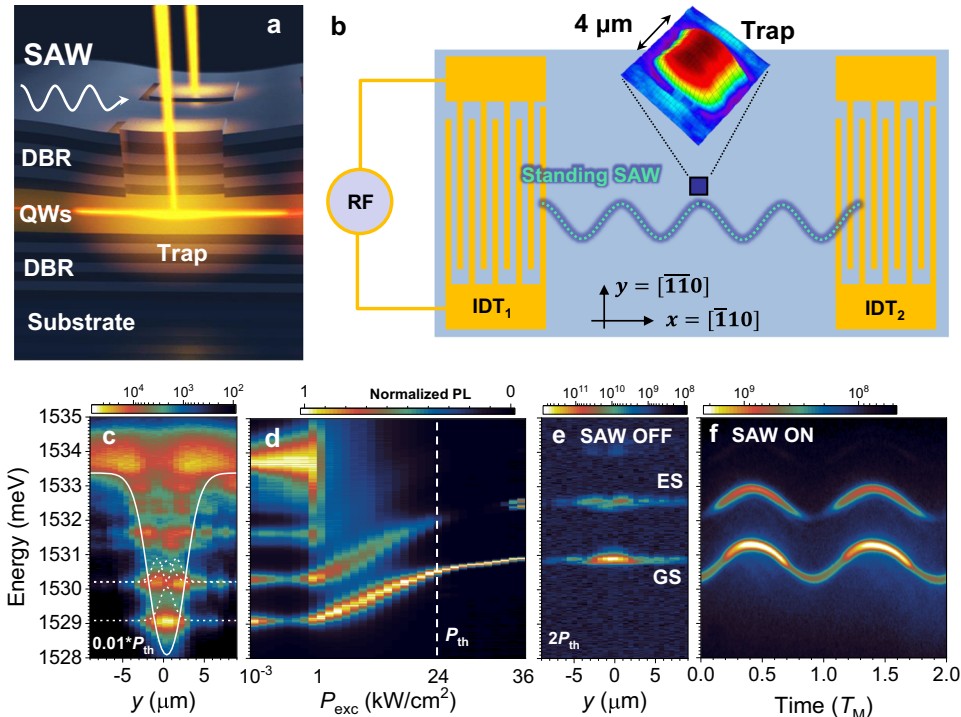

**Fig. 1 | Acoustically modulated confined polaritons. a** A sketch of an intracavity polariton trap in a structured (Al,Ga)As microcavity modulated by a surface acoustic wave (SAW). Multiple quantum wells (QWs) are positioned within the spacer between two distributed Bragg reflectros (DBRs). **b** A schematics of a $4 \times 4$ μm² trap at an anti-node of the standing acoustic field of a SAW resonator consisting of two interdigital transducers (IDT) designed for an acoustic wavelength of 8 μm. The standing SAW is excited by driving IDT₁ with radio-frequency (RF). The inset displays an atomic force micrograph of the sample surface imaging the trap topography. **c** Spatially resolved photoluminescence (PL) map of a polariton trap recorded under low optical excitation fluence $P_{exc}$ (i.e., below the condensation threshold, $P_{th}$) at 10 K. The solid and dashed curves display the trap confinement potential and squared wavefunctions (for the lower confined levels), respectively, determined as described in Supplementary Note 5. **d** Dependence of the spatially integrated PL on $P_{exc}$. The dashed line marks the BEC threshold power. **e** PL maps of the Bose-Einstein condensates (BECs) formed in ground (GS) and first excited state (ES) and **f** their time evolution under a SAW with frequency $f_M = 384$ MHz and nominal amplitude $A_M \propto 0.56\sqrt{W}$. The time scale in panel **f** is in units of the SAW period, $T_M = 1/f_M$.

amplitude[18]. The inset shows that $\chi^{(GS)}$ depends linearly on the SAW strain amplitude, which is $\propto A_M$. Since $\hbar\Omega_M < \gamma^{(s)}$, individual SAW sidebands are not resolved in the spectra.

The effects of a very large modulation amplitude on the confined GS and ES are illustrated by red dots the high-resolution spectra in the main panels of Fig. 3a and b, respectively. The SAW amplitude was increased, in this case, to yield $\chi^{(GS)} = 220$ corresponding to a maximum frequency shift $\chi^{(GS)}\Omega_M/(2\pi) = 84$ GHz (0.35 meV) approximately twice as large as the maximum shift in Fig. 2. We note that this modulation amplitude is still more than an order of magnitude smaller than the depth of the confinement potential of 5 meV [cf. Fig. 1c]. For comparison, both panels also display reference PL spectra recorded in the absence of acoustic excitation, which again reveal a single line with a spectral width $\gamma^{(s)}/\hbar = 0.6$ GHz.

The most remarkable features of the SAW-modulated PL spectra of Fig. 3a, b are the well-defined spectral oscillations near the extrema of the energy modulation amplitude. These oscillations are not present in the spectra recorded for lower modulation amplitudes of Fig. 2. As will be justified below, they correspond to the *acceleration beats* arising from coherent effects induced at very high driving amplitudes. The *acceleration beats* are observed for both the GS and ES of the trap, with no clear energy correlation between the emission of the two states. The beating frequency increases progressively towards the energy modulation extrema, where it reaches its maximum value $f_c \approx 4$ GHz.

### Temporal correlation studies
The high modulation amplitudes are expected to introduce temporal correlations in the emission at time scales much shorter that the SAW temporal period. We probed these correlations using the

interferometric setup depicted in Fig. 4a. Here, an interferogram [cf. inset of Fig. 4a] is created by superimposing a PL image of the trap emission with its time-delayed and spatially reversed counterpart. The temporal delay ($\tau$) is controlled by the different lengths of the interferometer arms. The response of interferometric setup is proportional to the continuous autocorrelation function $I_{ac}(\tau)$ of the PL emitted at two time intervals delayed by $\tau$, i.e.,

$$I_{ac}(\tau) = \frac{1}{4}\left\langle \mathcal{E}_{LP}^{(s)}(t) + \mathcal{E}_{LP}^{(s)}(\tau-t) \right\rangle^2, \qquad (1)$$

where $\mathcal{E}_{LP}^{(s)}(t)$ is the light electric field at time $t$ and $\langle \ldots \rangle$ means a time averaging. Spectrally resolved autocorrelation interferograms were then obtained by spectrally filtering the interferometric images with a resolution of 0.15 meV using a spectrograph. Figure 4b displays such an interferogram as the function of the delay $\tau$ under a modulation amplitude $\chi^{(GS)} = 44$. The right inset displays the corresponding spectrum obtained by integrating the emission along the $\tau$-axis showing a double-peak in the PL spectrum introduced by the acoustic modulation. Temporal oscillations, $I_{ac}(\tau)$, can be identified in Fig. 4b by spectrally integrating the interferograms within an energy range. Figure 4c displays such an autocorrelation trace obtained by integration over the region enclosed by the dashed rectangle in Fig. 4b. As will become clear later, the oscillations in $I_{ac}(\tau)$ are more pronounced in the energy region around the extrema of the modulation spectrum (as indicated by the dashed rectangle).

The autocorrelation trace $I_{ac}(\tau)$ of Fig. 4c consists of oscillatory features at two time scales: the first are the oscillations with the short period $2\pi/\omega_{LP,0}^{(GS)}$ displayed in Fig. 4d. These oscillations are associated

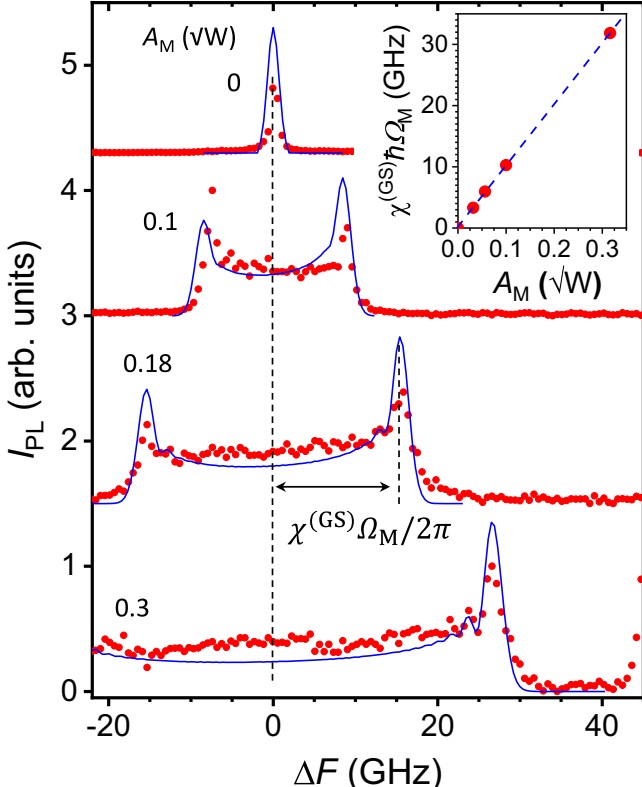

**Fig. 2 | Acoustically modulated polariton BECs.** High-resolution photo-luminescence spectra of the BEC ground state recorded for different values of $A_M = 0$, 0.1, 0.18, and 0.3 $\sqrt{W}$, which are proportional to the amplitude of the standing SAW strain field at the trap site. The PL intensity is normalized and spectra are shifted for clarity. The frequency shift $\Delta F$ is displayed relative to the zero-phonon line. The inset shows the energy modulation amplitude $\chi^{(GS)}\hbar\Omega_M$ as a function of $A_M$. The blue lines are fits to Eq. (6) using $\Delta M = 0.2$ and the $\chi^{(GS)}$ values indicated in the inset.

with the BEC zero-phonon line and thus, also present in the absence of acoustic modulation. The second type are the temporal beats (i.e., the amplitude modulation of the short-period oscillations) with an average periodicity $\bar{\tau}_{ac}$, which reduces with increasing modulation amplitude, as shown by the interferograms recorded for different $\chi^{(GS)}$ in Fig. 4e. The $\bar{\tau}_{ac}$ vs. $\chi^{(GS)}$ dependence summarized in Fig. 4f shows that the acoustic modulation can yield temporal correlations at time scales almost two orders of magnitude shorter than the modulation period $1/f_M$. These temporal beats thus provide direct evidence for the short-time correlations introduced by the high-amplitude acoustic modulation. The mechanisms behind these oscillations and their relationship to the *acceleration beats* will be discussed in the next section.

## Discussion

In order to understand the spectral changes induced by the energy modulation, we start by making some simplifying assumptions about the mechanisms for the interaction between the SAW and the polariton BEC. First, we assume that the SAW strain primarily modulates the bare excitonic resonances via the deformation potential (DP) mechanism[19], i.e., we neglect strain-induced radiation pressure effects, which, in polariton MCs are normally weaker in comparison to the DP effects. Furthermore, we will only consider effects associated with the hydrostatic component $s_h(t)$ of the SAW strain field, which will be taken to be homogeneous over the trap. Under these assumptions, the SAW-induced DP potential commutes with the polariton Hamiltonian and does not couple the confined states. If $s_h(t) = s_{h,0}\cos(\Omega_M t)$ denotes the SAW hydrostatic strain component, the energy of the bare exciton states in the QWs will then vary in time according to $\delta_M(t) = a_h s_h(t)$,

where $a_h$ denotes the exciton hydrostatic DP. By including the light-matter interaction with a coupling energy $\hbar\Omega_R$, it can be shown that the energy modulation of the $p^{th}$ confined polariton state can be expressed as (see, for details, Supplementary Note 2):

$$\hbar\omega_{LP}^{(s)}(t) = \hbar\omega_{LP,0}^{(s)} - \hbar\Omega_M\chi^{(s)}\cos(\Omega_M t), \quad (2)$$

with $\hbar\omega_{LP,0}^{(s)} = \frac{H_X^2-1}{2H_X^2-1}\delta_{CX}^{(s)}$ and $(\hbar\Omega_M)\chi^{(s)} = H_X^2 a_h s_{h,0}$. In these expressions, $\delta_{CX}^{(s)}$ denotes the energy detuning of the bare photonic level of state $s$ relative to the bare excitonic level. For the deformation potential interaction, $\chi^{(s)}$ becomes proportional to the (squared) exciton Hopfield coefficient $H_X^2$ [cf. Supplementary Eq. (3)] yielding the relative exciton contribution to the polariton wave function.

The linear dependence of the energy shifts $\hbar\omega_{LP}^{(s)}(t)$ on the relative modulation amplitude $\chi^{(s)}$ in Eq. (2) applies for modulation shifts small compared to the Rabi coupling $\Omega_R$. We show in Supplementary Note 2 that, for the conditions of the experiments, the frequency expansion in Eq. (2) taking into account only the first harmonic at $\hbar\Omega_M$ is a very good approximation over the whole range of SAW amplitudes investigated here. Finally, by assuming an hydrostatic deformation potential $a_h = -9$ eV for the GaAs QWs[20], we estimate from these equation a hydrostatic strain amplitude of $6 \times 10^{-5}$ for the modulation conditions of Fig. 3b.

The time evolution of the state wave function $\psi^{(s)}(t)$ then becomes:

$$\psi^{(s)}(t) = \psi^{(s)}(0)\exp\left[-\imath\int_0^t \omega_{LP}^{(s)}(\tau)d\tau\right] \quad (3)$$

$$= \psi^{(s)}(0)e^{-\imath\omega_{PL,0}t}e^{\imath\chi^{(s)}\sin(\Omega_M t)} \quad (4)$$

$$= \psi^{(s)}(0)e^{-\imath\omega_{PL,0}t}\sum_{n=-\infty}^{\infty}J_n(\chi^{(s)})e^{\imath n\Omega_M t} \quad (5)$$

from which one extracts the following expression for the amplitude of the emitted light field[21]:

$$\mathcal{E}_{LP}^{(s)}(t) = \mathcal{E}_{PL,0}\left[1 + \frac{1}{2}\Delta M_{PL}\cos(\Omega_M t)\right]$$
$$\times e^{-\imath\omega_{PL,0}t}\sum_{n=-\infty}^{\infty}J_n(\chi^{(s)})e^{\imath n\Omega_M t} \quad (6)$$

Here, the factor involving the amplitude modulation index $\Delta M$ (assumed to be $\ll 1$) is a phenomenological term included to account for changes in the total PL yield with modulation amplitude[21]. This index is defined as the maximum excursion in the PL intensity $I_{PL} \propto |\mathcal{E}_{LP}(t)|^2$ normalized to its time-averaged value.

The blue curves in Figs. 2 and 3 were determined by convoluting the sideband spectra obtained from to Eq. (6) with a Gaussian lineshape with a FWHM exactly equal to the one measured for the BEC in the absence of acoustic excitation in order to account for the finite temporal coherence of the BEC. Most of the spectral features are determined by the $\chi^{(s)}$. The $\Delta M$ merely introduces a reduction in the total emission intensity as well as an asymmetry in the intensity of the positive and negative sidebands (note that the term related to $\Delta M$ in Eq. (6) does not change sign under time reversal, while the others do). These results contrast with reports for electrically modulated vertical cavity lasers, where the sidebands are primarily induced by the $\Delta M$ factor[21].

The model reproduces remarkably well almost all spectral features of the GS, including an excellent fit of the *acceleration beats* at the extrema of the energy modulation of the GS [cf. right inset of Fig. 3a]. The same applies to the blue-shifted *acceleration beats* for the ES as illustrated by the right inset of Fig. 3b. In particular, we found that the amplitudes of the calculated *acceleration beats* are very sensitive to BEC temporal coherence. The excellent agreement between model and the experiments thus also proves that the acoustic pumping can

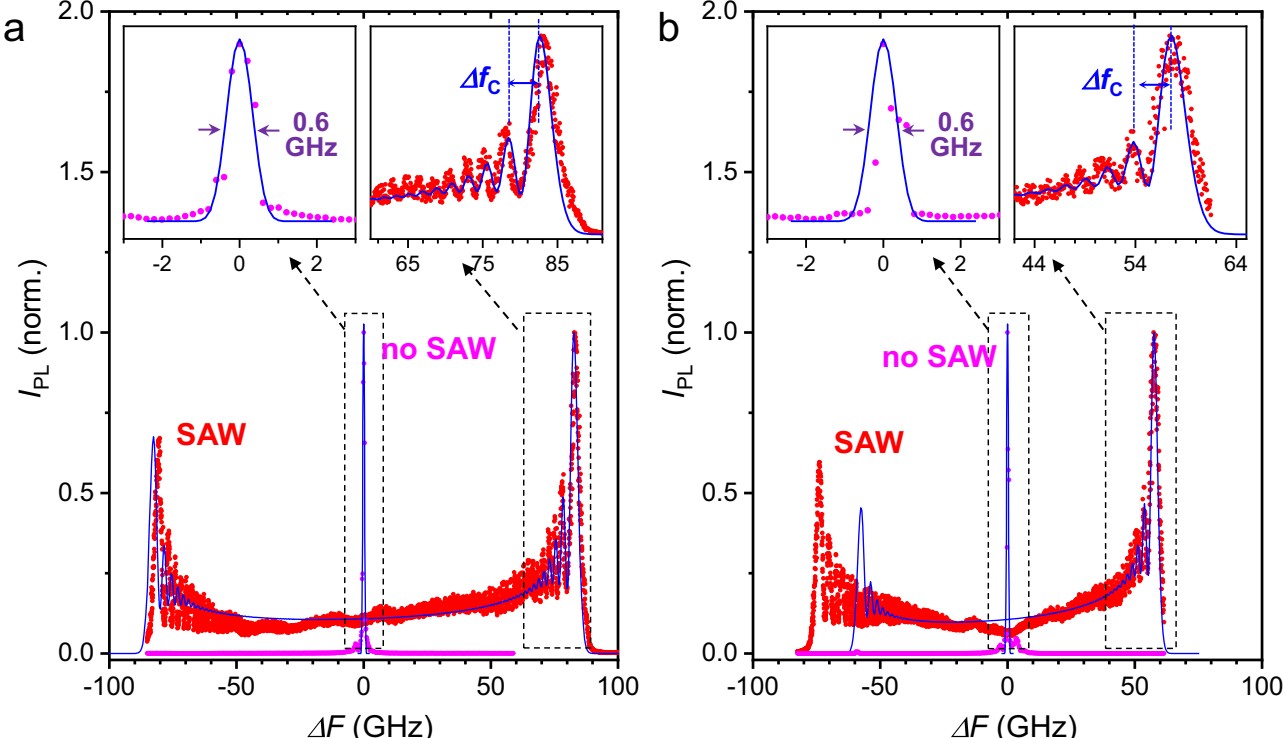

**Fig. 3 | Spectral *acceleration beats*.** High-resolution PL spectra of the BEC **a** ground state (GS) and **b** excited state (ES) spectra recorded for no SAW (magenta dots) and SAW amplitude $A_M = 0.56 \sqrt{W}$ (red dots) conditions. The blue lines are the fits to Eq. (6) yielding the parameters $\chi^{(GS)}\hbar\Omega_M = 84.4$ GHz and $\Delta M = 0.2$ for the GS and $\chi^{(ES)}\hbar\Omega_M = 59.1$ GHz, $\Delta M = 0.4$ for the ES. The raw PL data used to acquire the traces are displayed in Supplementary Fig. 1.

induce very large energy modulation amplitudes (e.g., $\chi^{(s)} > 200$) without introducing decoherence.

While the model predicts symmetric energy shifts for positive and negative sidebands, the negative energy shift for the ES is larger than the positive one [cf. Fig. 3b]. This asymmetry arises from quadratic terms in the dependence of the polariton energy shifts on the modulations amplitude, which are not taken into account by the linear dependence expressed by Eq. (2) and are more pronounced for the ES. The quadractic terms are attributed to the modulation of the trap barriers as well as to a spatially asymmetric distribution of the strain field in the trap location. If the polariton trap is exactly at an anti-node of the hydrostatic SAW strain field, this field has the least impact on the trap symmetry yielding almost equal positive and negative energy shifts. Slight shifts of the trap center relative to the anti-node can, however, induce large quadratic contributions to the energy modulation (cf. Supplementary Notes 2 and 5). These contributions depend on the symmetry and extension of the BEC wave function and lead to a strain-induced admixtures of confined levels[22], which are normally more pronounced for the ES than for the GS. We show in Supplementary Note 5 that shifts of the anti-node with respect to the traps center of less than 1 μm can already introduce the asymmetry observed for the ES in Fig. 3b while maintaining the GS energy shifts almost symmetric.

A deeper physical insight into the origin of the *acceleration beats* is obtained by expanding Eq. (4) for time intervals $t = t_m + \Delta t$ ($m = 0, 1, \ldots$) around the extrema of the modulation at $t_m = m\pi/\Omega_M$, which yields (see, for details, Supplementary Note 3):

$$\psi^{(s)}(t) = \psi^{(s)}(0)e^{-\iota\omega_{LP,0}^{(s)}t_m}$$
$$\times \exp\left[-\iota(-1)^m\chi^{(s)}\Omega_M\Delta t\right]\exp\left[\iota(-1)^m\frac{\chi^{(s)}}{6}(\Omega_M\Delta t)^3\right]. \quad (7)$$

If one neglects the $\Delta t^3$ term, a Fourier transformation of Eq. (7) yields sidebands $m\Omega_M$ with intensities proportional to $sinc^2[2\pi(m-\chi^{(s)})]$,

where $sinc(x) = \sin(x)/x$. Since $sinc(x)$ peaks at $x = 0$, one obtains a spectral shape similar to the ones in Fig. 2 with peaks at frequency shift $\Delta\omega_D = \pm\chi^{(s)}\hbar\Omega_M$. This emission pattern is analogous to the Doppler spectrum of a source emitting at $\hbar\omega_S$ at rest and moving with constant velocities $\mp c_0\Delta\omega_D/\omega_S$ relative to the observer, where $c_0$ is the speed of light.

The *acceleration beats* arise from the phase factors proportional to cubic and higher-order terms in $\Delta t$. While a closed-form expression for the Fourier transform of Eq. (7) cannot be obtained, we note that the $\Delta t^3$ term phase-modulates the Fourier components $e^{-\iota\Omega_M t}$ with a periodicity in time $\Delta t_c$ satisfying $\frac{\chi^{(s)}}{6}(\Omega_M\Delta t_c)^3 = 2\pi$. This temporal modulation induces a comb of frequencies with spacing $\Delta\omega_c = 2\pi\Delta f_c$ close to the extrema given by:

$$r_{ab} = \frac{\Delta\omega_c}{\Omega_M} = \frac{\Delta f_c}{f_M} = \sqrt[3]{\frac{2\pi^2}{3}\chi^{(s)}}. \quad (8)$$

Equation (8) implies that the observation of well-defined *acceleration beats* requires very large relative driving amplitudes $\chi^{(s)}$. As an example, if $\gamma^{(s)} \approx \hbar\Omega_M$, one typically needs $\frac{\Delta\omega_c}{\Omega_M} \gtrsim 10$ and $\chi^{(s)} \gtrsim 150$. This equation reproduces the frequency spacing $\Delta f/f_M = 11.2$ between the *acceleration beats* of Fig. 3b [upper left inset] very well using the fitted $\chi^{(GS)} = 220$. Note that the frequency spacing given by Eq. (8) applies for frequency shifts very close to the extrema: as one moves away, the spacing reduces together with the amplitude of the phase acceleration.

The large $\chi^{(s)}$'s required for the beats presuppose a strong interaction with the driving field, which must be introduced while maintaining temporal coherence and a pure harmonic character. In this respect, the use of a high modulation frequency is not always helpful since the accompanying high modulation amplitude $\chi^{(s)}\hbar\Omega_M$ may drive the system out of the harmonic regime. This situation is discussed in detail in Supplementary Note 4.

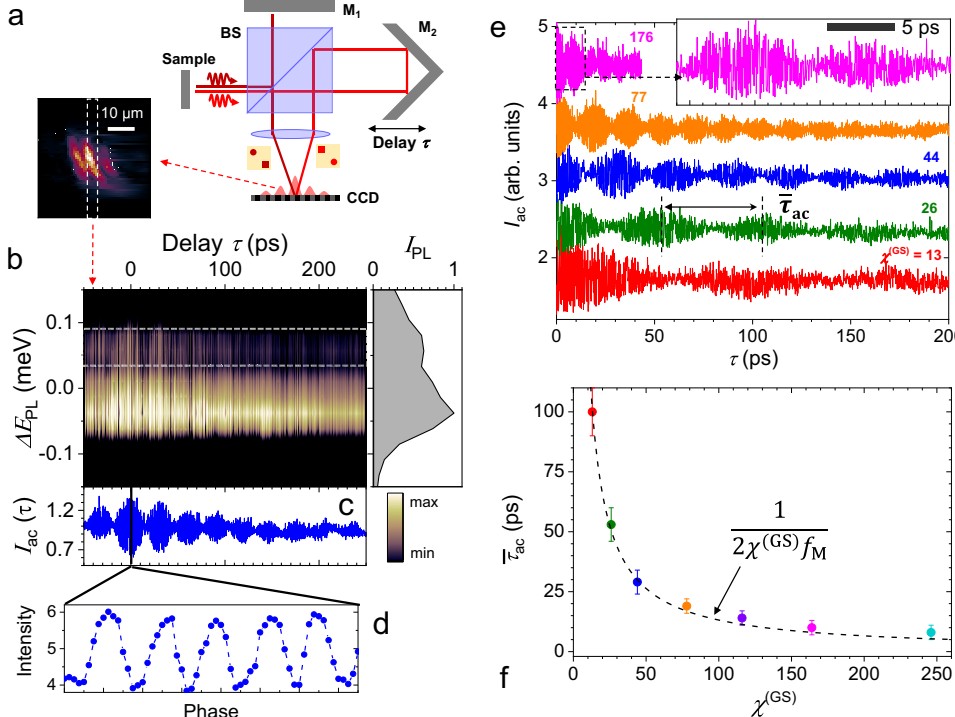

**Fig. 4 | Photoluminescence temporal correlations. a** Experimental setup for temporal correlation measurement under an acoustic field. An interferogram (inset on the lower left) is created by, first, splitting the signal on a 50:50 beamsplitter (BS) and, then, superimposing an image ($M_1$) of the PL with its time-delayed ($\tau$) and reversed counterpart produced by a retroreflector ($M_2$). This image is recorded by a CCD detector. **b** Energy-resolved interferogram for a modulation amplitude $\chi^{(GS)} = 44$ as the function of the delay $\tau$ obtained by spectrally filtering (resolution of 0.15 meV) the emission along the slit defined by the dashed line in the inset of panel a. The right inset displays the spectrum obtained by integrating the emission along the delay axis. The shaded background corresponds to the signal area. The energy axis is referenced to the GS BEC zero-phonon line at 1530.85 meV. **c** Interferogram $[I_{ac}(\tau, \chi^{(GS)})]$ obtained by integrating within the white horizontal dashed rectangle in panel (**b**). **d** Details of the oscillations in panel **c** with periodicity $2\pi/\omega_{LP,0}^{(GS)}$ associated with zero-phonon polariton line. **e** Interferograms, $I_{ac}(\tau, \chi^{(GS)})$, recorded for different amplitude modulation indices $\chi^{(GS)}$. The plots are normalized and displaced vertically for clarity. **f** Period $\bar{\tau}_{ac}$ of the oscillation beats in panel e as a function of $\chi^{(GS)}$ determined with the standard deviation indicated by the error bars. The dashed line corresponds to the dependence given by Eq. (11).

We now turn to the analysis of the time-correlation experiments and their relation to the high-resolution spectra. By substituting Eq. (6) into Eq. (1), one obtains the following expression for the autocorrelation function $I_{ac}(\tau)$:

$$I_{ac}^{(s)}(\tau, \chi^{(s)}) = \left\langle \mathcal{E}_{LP}^{(s)}(t)\mathcal{E}_{LP}^{(s)}(\tau - t) \right\rangle = \cos(\omega_{PL,0}\tau) \sum_{m=-\infty}^{\infty} I_{m,ac}^{(s)}\tau; \qquad (9)$$

where

$$I_{m,ac}^{(s)}(\tau, \chi^{(s)}) = e^{-\frac{\gamma^{(s)}}{2}\tau}J_m^2[\chi^{(s)}]\cos(m\Omega_M\tau) \qquad (10)$$

In these expressions, the intensity modulation index $\Delta M$ was assumed to be zero. The autocorrelation function $I_{ac}^{(s)}(\tau, \chi^{(s)})$ is thus simply a sum of the individual contributions of the spectral autocorrelation functions $I_{m,ac}^{(s)}(\tau, \chi^{(s)})$ associated with the sideband of order $m$ and angular frequency $m\Omega_M$.

Figure 5a displays $I_{m,ac}^{(GS)}(\tau, \chi^{(GS)})$ as a function of $\tau$ (horizontal scale) and the sideband index $m$ (vertical scale) calculated for a relative modulation amplitude $\chi^{(GS)} = 77$ [corresponding to the orange curve in Fig. 4e]. For each $m$, the periodicity of the time oscillations is given by $2\pi/(|m|\Omega_M)$ [cf. Eq. (10)], while their decay within the displayed $\tau$ range is negligible since $\tau\gamma^{(GS)} \ll 1$. In contrast, when $I_{ac}^{(s)}(\tau, \chi^{(s)})$ is integrated over an energy range [such as the rectangle area in Fig. 4b and the range of $m$'s denoted by the asterisk in Fig. 5a], the superposition of oscillations with different periods induces an amplitude decay of the autocorrelation traces at time scales much shorter than $\gamma^{(GS)}$, which reduces with $\chi^{(GS)}$. This mechanism accounts for the relatively small decay range of the oscillations at high $\chi^{(GS)}$'s in Fig. 4e.

The amplitude of the oscillations is small except for the components around $m = \pm\chi^{(GS)}$. As a consequence, the periodicity of the oscillation beats of the integrated correlation function $I_{ac}^{(s)}(\tau, \chi^{(s)})$ becomes approximately equal to

$$\bar{\tau}_{ac} = 1/(2\chi^{(GS)}f_M). \qquad (11)$$

Here, the factor of 2 arises from the fact that the fast-changing $\cos(\omega_{PL,0}\tau)$ term in Eq. (9) makes the periodicity of the envelope of $I_{ac}^{(s)}(\tau, \chi^{(s)})$ half of the one of $I_{m,ac}^{(s)}(\tau, \chi^{(s)})$.

The $\bar{\tau}_{ac}$ values calculated from the previous equation are in excellent agreement with the experimental results plotted in Fig. 4f. Furthermore, correlation profiles $I_{ac}^{(GS)}(\tau, \chi^{(GS)})$ calculated from Eq. (9) for different $\chi^{(GS)}$ and displayed in Fig. 5b reproduce well the corresponding experimental features in Fig. 4e.

Finally, a remaining question is the impact of the *acceleration beats* on the temporal correlation function. These beats appear in Fig. 5a as a sequence of resonances shifted from the modulation extrema at the energy $\chi^{(s)}\hbar\Omega_M$ by multiples of approximately $r_{ab}\hbar\Omega_M$ [cf. Eq. (8)]: the corresponding range of sideband indices are indicated by a "*" in the figure. The interference of these modes gives rise to correlation envelopes with periodicity given by $\tau_{ab}^- \approx \pi/[(\chi^{(s)} - r_{ab})\Omega_M]$ and $\tau_{ab}^+ \approx \pi/[(r_{ab})\Omega_M]$. The effect of the longer periodicity $\tau_{ab}^+$, which is associated with the energy splitting between accelerations beats, is indicated in the simulated spectra for $\chi^{(GS)} = 176$ and $\chi^{(GS)} = 77$ in Fig. 5b: it induces small modulations with periodicities $\tau_{ab}^+ \approx 150$ ps and 250 ps, respectively, which are much larger than $\bar{\tau}_{ac}$.

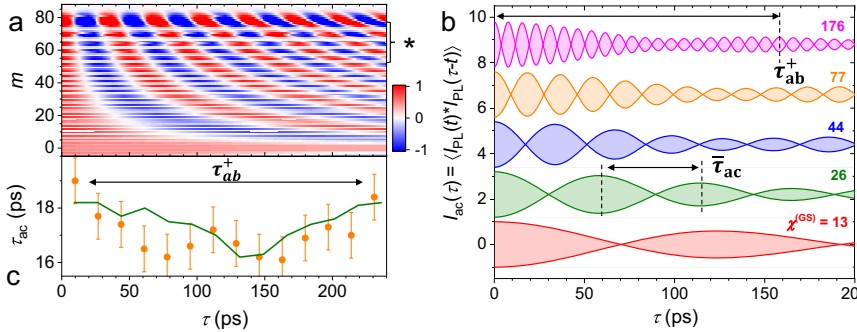

**Fig. 5 | Photoluminescence temporal correlations. a** Spectral autocorrelation function $I_{m,ac}^{(GS)}(\tau, \chi^{(GS)})$ [cf. Eq. (10)] calculated for $\chi^{(GS)} = 77$. The vertical axis represents the sideband index, $m$ (note that $I_{m,ac}^{(GS)}(\tau, \chi^{(GS)})$ is symmetric with respect to $m = 0$). **b** Interferograms $I_{m,ac}^{(GS)}(\tau, \chi^{(GH)})$ [cf. Eq. (9)] calculated for different $\chi^{(GS)}$. The shaded regions designate the signal area. **c** Orange circles: experimental values for $\tau_{ac}$ extracted from Fig. 4e as a function of the delay $\tau$ with the standard deviation indicated by the error bars. The green line represents values obtained from the simulation in panel a by integrating the emission over the range of $\chi$ denoted by an asterisk.

The amplitude of the autocorrelation beats associated with $\tau_{ab}^{+}$ depends on the amplitude ratio between the *acceleration beats* close to the extrema of the modulation. This amplitude is small and, due to the limited delay range and signal-to-noise ratio of the experiments, the intensity envelopes associated with $\tau_{ab}^{+}$ can not be clearly identified in the experimental curves of Fig. 4e. We note, however, that in addition to the intensity envelopes, the mode interference also induces small fluctuations in the spacing $\tau_{ac}$ between the time-domain beats. Qualitatively, these fluctuations can be understood by considering the autocorrelation traces $I_{m,ac}^{(GS)}(\tau, \chi^{(GS)})$ close to $m = \chi^{(GS)}$ in Fig. 5a: with increasing delay $\tau$, the lower-$m$ intensity beats with larger repetition period dephase from the strongest one at $m = \chi^{(GS)}$. When one integrates the emission over a range of energy, the spacing $\tau_{ac}$ between the beats oscillates within the range $\tau_{ab}^{-} \leq \tau_{ac} \leq 1/(2\chi^{(GS)} f_M)$, which repeats with a periodicity $\tau_{ab}^{+}$. The solid circles in Fig. 5c display the $\tau_{ac}$ values extracted from the measured autocorrelation profile of Fig. 4e for $\chi^{(GS)} = 77$. The calculated dependence of $\tau_{ac}(\tau)$ for the same $\chi^{(GS)}$, given by the solid line, matches very well the experimental data. Although not as clear as in the spectral domain (cf. Fig. 3), this period reduction gives evidence for the *acceleration beats* in the time-domain.

In conclusion, we have presented experimental evidence for *acceleration beats* in harmonically driven oscillators, which are novel spectral features induced by accelerated (i.e., quadratic or higher order in time) energy changing rates at high driving amplitudes. Optomechanical systems using high-frequency surface acoustic waves enable us to reach the high acceleration regime by providing large modulation amplitudes while preserving the system's coherence. For small driving amplitudes, the main impact of the acceleration is a modulation of the sideband amplitudes. At high driving, the acceleration terms induce temporal correlation at a time scale much shorter than the driving period, which lead to frequency beats in the spectral response. The large beating frequencies relax the coherence requirements for the observation of non-adiabatic behavior. Finally, the *acceleration beats* are a general feature of systems with accelerated frequency changes: they are thus also expected for accelerated radiation sources, e.g., those under strong gravitational gradients.

## Methods
### Intracavity polariton traps
The intracavity traps depicted in Fig. 1a and b were defined within the spacer layer of an (Al,Ga)As MC (quality factor of approx. 5000 and a Rabi splitting $2\hbar\Omega_R = 7.2$ meV, where $\Omega_R$ is the Rabi coupling) on GaAs (001) by combining growth steps by molecular beam epitaxy (MBE) with patterning[17,23,24]. The lower distributed Bragg reflector (DBR) and the MC spacer region containing three pairs of 15 nm thick GaAs quantum wells (QWs) were deposited in the first MBE growth step. The sample was then removed from the MBE chamber and patterned by means of photolithography and wet chemical etching to form approx. 12 nm high mesa with μm-sized lateral dimensions and then reinserted in the MBE chamber for the deposition of the upper DBR[22]. The etching produces traps with an optical confinement energy of a few meV in the mesa regions.

### Surface acoustic wave (SAW) resonators
The acoustic delay line for SAWs consisting of two interdigital acoustic transducers (IDTs) was designed for a SAW wavelength $\lambda_M = 8$ μm. This relatively long acoustic wavelength ensures that the evanescent SAW field penetrates the upper DBR and reaches the MC spacer containing the QWs. The IDT fingers consisting of a Ti/Al/Ti stack of thicknesses of 10/30/10 nm were defined by optical lithography and metallization lift-off. One of the IDTs of the delay line was driven by an radio frequency around 384 MHz leading to the excitation of a SAW of the same frequency, while the second one acts as an acoustic reflector, leading to the formation of an acoustic cavity.

### Photoluminescence spectroscopy
The spectroscopic photoluminescence (PL) studies of the acoustic modulation were performed at low temperatures (10 K) on an intracavity polariton square trap with nominal dimensions of $4 \times 4$ μm². The confinement potential of the studied trap was given by the energies of the more photonic lower polariton inside the mesa (bottom) and more excitonic lower polariton outside (barrier), leading to the confinement energy of approx. 5 meV.

Polaritons in the trap were non-resonantly excited by a 780 nm continuous wave beam of a cavity-stabilized Ti-sapphire laser focused onto a 30 μm spot on the sample surface. The time-integrated emission from the confined states was spectrally analyzed by a spectrometer and then detected by an liquid-nitrogen-cooled CCD camera[16]. The time-resolved PL detection of the emission of confined polaritons under acoustic modulation was performed by using a streak-camera synchronized with the SAW frequency. The spectroscopic investigations with high spectral resolution were carried out by inserting an etalon (resolution of 300 MHz and free spectral range 68 GHz) before the spectrometer. Further details about the experimental setup can be found in Supplementary Note 1.

Time-correlation measurements were carried out using a Michelson interferometer with a retro-reflector [$M_2$, Fig. 4a] placed on a motorized delay line. The BEC PL image interferes with its spatially inverted and time-delayed counterpart on the entrance slit of the spectrometer. One of the fringes of the resulting interferogram was selected by the slit for spectral analysis. For the autocorrelation measurements of Fig. 4c, e, only the retro-reflector was moved, while the mirror in the other arm was kept fixed. In the case of the measurement

of Fig. 4d, the retro-reflector arm was fixed, while the other mirror ($M_1$) was scanned with nm-scale steps by a piezo-motor to measure the evolution of the BEC phase over a few oscillation periods.

## Data availability

The measurement and numerical simulation data that support the findings within this study are included within the main text and Supplementary Information and can also be made available upon a request from the corresponding author.

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

## Acknowledgements

We acknowledge the funding from German DFG (grants 359162958 and 426728819). The authors thank A. Pitanti and M. Msall for discussions and for a critical review of the manuscript as well as the technical support by R. Baumann, S. Rauwerdink, W. Anders, and A. Tahraoui.

## Author contributions

A.S.K. performed the experiments, K.B. carried out the MBE growth of the samples, P.V.S. developed the model for the *acceleration beats*. A.S.K. and P.V.S. have conceived the idea, analyzed the data and wrote the paper.

## Funding

## Competing interests

The authors declare no competing interests.
