## [Peer Review File · Nature Communications]

Acceleration-induced spectral beats in strongly driven harmonic oscillatorsREVIEWER COMMENTS

Reviewer #1 (Remarks to the Author):

The paper reports on clear experimental observation of frequency combs both in adiabatic and non-adiabatic regimes under high frequency harmonic modulation of confined polariton condensate system. Convincing theoretical model explaining the observed results is also presented. However the novelty of the observations is partially masked by earlier work Phys Rev B 80, 075301 (2009) in which similar sidebands (acceleration beats) are reported in semiconductor microcavity under ultrafast strain pulses. In this respect the underlying phenomena are similar with the main difference being that the authors report such sidebands in the regime of trapped condensation and perfectly harmonic modulation, whereas earlier work observed them in reflectivity spectra under weak excitation regime below threshold using single strain pulse. To conclude, I believe the reported results are solid and their theoretical interpretation sound, however due to the above issue regarding the novelty of the observed phenomena I would recommend the publication of the paper in a more specialized journal. Several points need to be considered before the paper is published.

- 1) The abstract mainly discusses the theoretical background with only one sentence addressing the actual experimental achievements. In my opinion the abstract should be rewritten to emphasize more the reported work.
- 2) It would strongly add to the strength of the paper if the authors could present direct measurements of temporal correlations on time scale shorter than the modulation period. Presently this is implied only from the observed spectral features.
- 2) There are several typo errors for example in line 12, 128.

Reviewer #2 (Remarks to the Author):

In their manuscript, the authors experimentally demonstrate a new regime of non-adiabaticity by studying the spectral response of a polariton BEC subject to a high amplitude surface acoustic wave (SAW) driving field. For a SAW driving frequency lower than the polariton decoherence rate, they observe a transition to the non-adiabatic regime as the amplitude of the SAW is increased. The emission spectrum of the polariton BEC develops new spectral modulations, termed acceleration beats, which are well reproduced by the theoretical model developed in the manuscript.

Overall, the manuscript is well written and insightful, and could be of interest to a diverse community of readers as the physics explored is not restricted to the polariton BEC platform. The main claims are well supported by the experimental data and simulations. I would recommend the manuscript for publication, after the authors address the following comments:

1- The behavior of the polariton BEC in Fig.1e,f is not obvious and would deserve more detailed description in the main text. Why does BEC occurs both in the ground state and in the 1st excited state? Is the relative population dictated by a thermal occupation factor? Does the population ratio between these two states indeed remain the same in the presence of strong SAW driving? In Fig1f, the period of the spectral oscillations appear to be ~ 1 ns instead of the 2.6ns period of the SAW field; why is this the case? How well does the model reproduce the temporal variations of emission energy and emission intensity shown in Fig.1f?

2- The 2D representation chosen in Fig3a is hard to decipher, and it is not clear how the spectra of the other panels are derived from such maps. Were the data of Fig2 acquired in the same way? If so, the same choice of representation should be kept in both figures, in order to make it clear that only the SAW amplitude is being varied. From this figure, it appears that the integrated emission from the BEC excited state is higher than that of the ground state, contrary to what is shown in Fig1. Is this an artifact of the measurement technique, or is their more to it?

3- In Fig.3c, the data and simulation are quite off for the negative energy shift regime. It is argued in

the main text that this discrepancy could be attributed to strain-induced admixture between the BEC excited state and other confined levels. First, it is not clear from Fig1 that such higher order confined levels still exist in the trap at the pump fluence used in the experiment due to the mean-field energy shift of the polariton BEC. Second, the authors should justify why such an effect would only affect the negative energy shift part of the spectrum.

4- The discussion of the theoretical model for the interaction between SAWs and the polariton BEC is too short and technical for a reader who is not expert in the field. In particular, the simplifications made regarding the strain-induced radiation pressure effects and the hydrostatic component should be given more justifications in order for the reader to understand the physical assumption of the model used in the manuscript, without having to dive into the SAW literature.

5- Lines 198 and 205, "Eq.(7)" should be replaced by "Eq.(6)"

Reviewer #3 (Remarks to the Author):

This paper reports the observation of interference effects in the optical emission spectrum induced by the rapid energy modulation applied to polariton BEC states. The modulation is induced by the periodic acoustic strain through SAW excitation and the phase delay/advance caused by the frequency acceleration between two time-durations sandwiching the highest strain point causes the destructive and constructive interference, which leads to the periodic PL intensity modulation near the spectral band edges. The physics behind is clear and interesting, and it is nice to see the effect of temporal coherence even though the polariton decoherence time is shorter than the period of strain modulation, so that I believe it can give a new universal insight into the hybrid systems consisting of highly frequency-mismatched systems. However, there are points to be clarified and I recommend the authors to improve the manuscript before recommending the publication.

1. The role of phase decoherence is not clear. If I correctly understand, the theoretical framework is first based on the perfectly coherent system and the effect of decoherence is then introduced simply by using the Gaussian smoothing. I believe that this is very rough discussion because the suppression of interference is not caused by something like inhomogeneous broadening but by the phase decoherence. In fact, I guess that the observed interference is suppressed also when the phase decoherence time is much larger than the SAW period. This is because the incommensurate relation between the SAW and PL frequencies causes different phase shift between the two time-durations with a similar emission energy but separated longer than one period. (Of course, side bands start to appear in this situation, but the averaging should show the similar trend with suppressed interference.) Therefore, the interference pattern is observed only within the case when the decoherence time is shorter than the period but larger than the time required to induce the interference. Then, I wonder if simply using the Gaussian smoothing is not suitable for describing this system. I recommend the authors to describe the model based on the phase decoherence and clarify the condition to observe the effects. (They may be performed finite-time integration in Fourier transformation, but it is not explicitly mentioned.)

2. The onset of modulation amplitude to observe this phenomenon gives a quantitative knowledge on the decoherence time. If the author can include this discussion, it would be more interesting and useful. Probably, the model for discussing the phase decoherence is required.

3. The SAW modulation for the energy shift of $\sim 100\text{GHz}$ is large and comparable to the polariton trapping potential, possibly inducing the nonlinear energy shift. Does this cause additional effects?

4. The role of BEC is unclear. It is required to have enough long decoherence time?

5. There are many typos. The author should much more carefully prepare the manuscript. For examples,

- Many index (s) depicting the polariton mode is replaced by (p): Line 158, 160, 345, 346, 3rd line above 354, 354, etc. Both can be used by should be unified.

- Duplicate words: "again" in line 128, "using the" in 213, "potential" in 336.

- Numbering of Figures: Use ones of 3a, 3b, 3c or 3(a), 3(b), 3(c) for example in page 7.
- Do you need two π in line 160?
- In many parts, energy and frequency is confusingly used. Insert or delete \hbar correctly. For example, Fig 2(c), line 160, 201, SM1, SM2, below 353, 354,
- In line 196, m is missing.
- Line 198, Eq.(7) should be (5) or (6)?
- small π in line 372.
- Fig. SM2 caption: (a) $0.2 \rightarrow 0$ and (b) the value of ΔM_{PL} should be shown (maybe 0.2 ?).
- Schrodinger equation SM5 does not include the wavefunction in R.H.S.

Response to Reviewers.

Reviewer 1 remarks

"The paper reports on clear experimental observation of frequency combs both in adiabatic and non-adiabatic regimes under high frequency harmonic modulation of confined polariton condensate system. Convincing theoretical model explaining the observed results is also presented".

Point 1.1:

"However the novelty of the observations is partially masked by earlier work Phys Rev B 80, 075301 (2009) in which similar sidebands (acceleration beats) are reported in semiconductor microcavity under ultrafast strain pulses. In this respect the underlying phenomena are similar with the main difference being that the authors report such sidebands in the regime of trapped condensation and perfectly harmonic modulation, whereas earlier work observed them in reflectivity spectra under weak excitation regime below threshold using single strain pulse. To conclude, I believe the reported results are solid and their theoretical interpretation sound, however due to the above issue regarding the novelty of the observed phenomena I would recommend the publication of the paper in a more specialized journal. Several points need to be considered before the paper is published."

Response:

We thank the reviewer for these observations. On the one side, we disagree with the referee's assertion that the "acceleration beats" reported here are "similar" to the non-adiabatic sidebands reported by *Berstermann et al.* (Ref. [12]), as will be justified below. On the other side, the referee's observations have made clear to us that the physical novelty reported in the paper, the experimental evidence for acceleration beats as well as the difference between these beats and non-adiabatic sidebands, were not clearly highlighted in the first version of the manuscript. This comment gives us now the opportunity to better clarify the major differences between our results and the conventional non-adiabatic sidebands previously reported in that reference as well as in other works.

We were aware of the impressive results of fast strain modulation from Ref. [12]. They were not mentioned in the original submission because the modulation is not harmonic and, as we will show below that, the main features discussed in this reference are associated with conventional non-adiabatic sidebands, which are qualitatively

distinct from the "acceleration beats" reported here. The analysis of the experimental data in Ref. [12] focus on the conventional non-adiabatic modulation regime "*induced on time scales shorter than the polariton decoherence*", leading to the "*characteristic sidebands, which are spectral fingerprints of the terahertz modulation process appear in the spectrum near the polariton resonance*". The discussion explicitly mentions only the regime in which the "*resonance energy changes linearly with time*", i.e., "velocity-related" changes following a time dependence t^m with $m = 1$ (all quotes above are from the reference). This regime contrasts with the phenomenology discussed in our manuscript – the *acceleration beats*, which arise from accelerated changes in the evolution of the energy with time (i.e., following a time dependence t^m with an integer $m > 1$). These "acceleration" effects are not discussed in Ref. [12].

In order to clarify these issues, we made the following modifications:

- The "acceleration" regime has probably not been considered in Ref. [12], in part because the experimental conditions were not appropriate to distinguish the "acceleration"- related changes from the dominating ones due to "velocity"-related changes. In order to highlight the difference, we included a new section in the supplement (Sec. SM4. Conditions for the observation of acceleration beats) to highlight the challenges for detecting acceleration beats as well as to show that the main spectral features in Ref. [12] arise from velocity-induced changes. Due to the concerns of Reviewer 1, we list below this section:

The main challenge for the identification of features due to "acceleration" is to discriminate them from the dominating ones related to linear energy changing rates associated with "velocity" (or Doppler) effects. In the case of harmonic modulation, we show in the manuscript that this can be achieved by increasing the energy modulation index, χ , while preserving the harmonic character of the modulation. In this respect, the use of a high modulation frequency $f_M = \Omega_M/(2\pi)$ is not always helpful since the accompanying high modulation amplitude $\chi\hbar\Omega_M$ may drive the system out of the harmonic range and introduce significant harmonic distortion, which complicates the identification of acceleration effects.

A few publications have reported modulation procedures yielding a large number of sidebands in different physical systems (see, e.g., Refs. [3, 4, 27]). To our knowledge, however, the conditions for the observation of clear acceleration beats were not achieved in any of these reports. As an example, we consider the modulation of polaritons reported in Ref. [12], which is one of the few examples involving polaritons. The strain modulation in this case was carried out using very fast strain pulses and is, thus, not harmonic. The most impressive result is the demonstration of non-adiabatic sidebands for modulation in the sub-BEC threshold regime. The modulation is fast enough to yield well-defined sidebands even for polariton coherences corresponding to a linewidth of 0.18 meV, significantly larger than those reported here for polariton BECs.

Despite the very fast modulation, the main spectral features are still due to "velocity"-related changes rather than to "acceleration" since the requirements listed above for the observation of acceleration beats are not satisfied. To prove this assertion, we made a rough estimate of the relative ratios between the linear and quadratic contributions by approximating the strain modulation (cf. Fig. 1(e) of the main text in Ref. [12]), which consists of on a few (3) sine-like cycles, by a sinusoidal one with the same amplitude (0.37 meV, the maximum lower polariton shift mentioned in the reference) and an effective frequency of 40 GHz (0.16 meV). The latter yields an effective modulation index of only $\chi = 2.3$. The sideband spectrum determined from Eq. (6) in the main text for an harmonic modulation with $\chi = 2.3$ is displayed in Fig. SM6: one obtains only three well-defined sidebands spaced by $\hbar\Omega_M$, in good agreement with the experimental results presented in Fig. 2b of the reference paper. These are conventional sidebands related to the "velocity" energy changes expressed by Eq. (SM8), which take place over a substantial fraction of the harmonic modulation cycle: here, the red- and blue-shifted sidebands appear, respectively, for positive (i.e., away from) and negative (towards) "velocities" relative to the observer.

The very small number of sidebands in Fig. SM6 makes it impossible to see the bunching effects associated with *acceleration beats*, even in the case of a perfect harmonic modulation. In contrast with these results, our data in Fig. 3 of the main text was recorded for a comparable energy modulation amplitude (of approx. 0.35 meV) but at a much lower SAW frequency, leading to energy modulation indices χ values two order of magnitude larger. In the harmonic regime under long temporal coherences, it is a large modulation index χ that enables pronounced acceleration beats rather than a high frequency or energy modulation amplitude.

Finally, the modulation by the SAW in Fig. 3 of the main text remains harmonic (and monochromatic) up to very high χ 's, which enables the description of the "acceleration effects" using the simple expression in Eq. (6). That is not the case for the pulsed excitation process used in Ref. [12]. As an

FIG. 1. *

Fig. SM6 Sideband spectrum for the harmonic modulation with frequency $f_M = 40$ GHz and amplitude $\chi\hbar f_M$ with $\chi = 2.3$ as determined from Eq. (6) of the main text.

example, the authors mention a nonlinear decrease in the sideband spacing with strain amplitudes, which they attributed to nonlinear pulse propagation effects.

To emphasize the different nature of the findings from Ref. [12], the following modifications were carried out in the revised main text:

- We first make a clear distinction between conventional non-adiabatic sidebands (related to velocity) and acceleration beats by rephrasing the last sentences of the 3rd paragraph of page 3:
Contrary to the conventional sidebands observed for small harmonic modulation amplitudes $\chi^{(s)}$, which arise from energy change rates linear in time and have a fixed energy separation $\hbar\Omega_M$, the *acceleration beats* arise from the bunching of sidebands to form combs with an energy spacing that increases with $\chi^{(s)}$. This spacing can exceed the state decoherence, thus leading to a non-adiabatic response even for $\hbar\Omega_M < \gamma^{(s)}$.
- In order to clarify the challenges associated with the detection of acceleration beats in the main text, we modified the last sentence of the last paragraph of page 3 to read:
Here, one of the reasons is that their observation requires large energy modulation amplitudes relative both to the modulation quantum and the decoherence rate, which must be achieved while maintaining the temporal coherence and the harmonic character.
- The following sentences were added to the second paragraph of page 14:
The large $\chi^{(s)}$'s required for the beats presuppose a strong interaction with the driving field, which must be introduced while maintaining temporal coherence and a pure harmonic character. In this respect, the use of a high modulation frequency is not always helpful since the accompanying high modulation amplitude $\chi^{(s)}\hbar\Omega_M$ may drive the system out of the harmonic regime. This situation is discussed in detail in Sec. SM4.
- Reference for the work of Berstermann et al. has now been included by modifying the following sentence of the second paragraph of page 4:
These waves have been applied to generate non-adiabatic sidebands in long-living atomic-like states such as those in single color centers [5, 6] and quantum dots [7–11]. Non-adiabatic modulation by fast strain fields has also been reported for microcavity polaritons [12] as well as for the confined microcavity polariton condensates investigated here [13, 14].

Point 1.2:

"The abstract mainly discusses the theoretical background with only one sentence addressing the actual experimental achievements. In my opinion the abstract should be rewritten to emphasize more the reported work."

Response:

We have modified the abstract to better describe the experimental results presented in the paper (while complying to the length limit established by the journal). The updated abstract is presented below.

The harmonic modulation of coherent systems gives rise to a wealth of physical phenomena, e.g., the AC-Stark effect and Mollow triplets, with important implications for coherent control and frequency conversion. Here, we demonstrate a novel regime of temporal coherence in oscillators harmonically driven at extreme energy modulation amplitudes relative to the modulation quantum. The studies were carried out by modulating a confined exciton-polariton Bose-Einstein condensate (BEC) by an acoustic wave. Features of the new regime are the appearance, in the spectral domain, of a comb of resonances termed *acceleration beats* with energy spacing tunable by the modulation amplitude and, in the time domain, of temporal correlations at time scales much shorter than the acoustic period, which also depend on the modulation amplitude. These features are quantitatively accounted for by a theoretical framework, which associates the *beats* with accelerated energy-change rates during the harmonic cycle. These observations are underpinned by the high sensitivity of the BEC energy to the acoustic driving, which simultaneously preserves BEC's temporal coherence. The acceleration beats are a general feature associated with accelerated energy changes: analogous features are thus also expected to appear under highly accelerated motion e.g., in connection with Cherenkov and Hawking radiation.

Point 1.3:

"It would strongly add to the strength of the paper if the authors could present direct measurements of temporal correlations on time scale shorter than the modulation period. Presently this is implied only from the observed spectral features."

Response:

We thank the referee for this insightful suggestion. Indeed, measurements of the temporal coherence on a time scale shorter than the SAW period provide a complementary approach to detect non-adiabatic effects arising from the high harmonic modulation amplitudes. We have carried out time-correlation measurements with ps-resolution under strong acoustic excitation. In summary, for high modulation amplitudes, the temporal correlation profiles reveal two periodicities with time scales much smaller than the SAW frequency, which, via a comparison with the simulations, are shown to be associated with the acceleration beats.

In order to address this point, we introduced the following major changes (since the text is long, the modifications are only summarized here):

- Section II (Results) was extended to include the direct temporal correlation experiments suggested by the referee (pages 8-11). A new figure was introduced (Fig. 4) summarizing these results.
- Sec III (Discussions and Conclusions) was extended to include the analysis of the temporal correlation data as well as relationship to the acceleration beats in the frequency domain as well as a new figure (Fig. 5) (pages 14-16).

Point 1.4:

"There are several typo errors for example in line 12, 128."

Response:

We thank the referee for pointing out these typos, which have been corrected in the revised version.

Reviewer 2 remarks

"In their manuscript, the authors experimentally demonstrate a new regime of non-adiabaticity by studying the spectral response of a polariton BEC subject to a high amplitude surface acoustic wave (SAW) driving field. For a SAW driving frequency lower than the polariton decoherence rate, they observe a transition to the non-adiabatic regime as the amplitude of the SAW is increased. The emission spectrum of the polariton BEC develops new spectral modulations, termed acceleration beats, which are well reproduced by the theoretical model developed in the manuscript.

Overall, the manuscript is well written and insightful, and could be of interest to a diverse community of readers as the physics explored is not restricted to the polariton BEC platform. The main claims are well supported by the experimental data and simulations. I would recommend the manuscript for publication, after the authors address the following comments:"

Point 2.1:

” The behavior of the polariton BEC in Fig.1e,f is not obvious and would deserve more detailed description in the main text. Why does BEC occurs both in the ground state and in the 1st excited state? Is the relative population dictated by a thermal occupation factor? Does the population ratio between these two states indeed remain the same in the presence of strong SAW driving? In Fig1f, the period of the spectral oscillations appear to be 1ns instead of the 2.6ns period of the SAW field; why is this the case? How well does the model reproduce the temporal variations of emission energy and emission intensity shown in Fig.1f?”

Response:

The polariton ”BEC” is a non-equilibrium state (and thus different from a real BEC in thermodynamic equilibrium): it is maintained by compensating particle losses by photon escape from the microcavity via the stimulated scattering from polariton states excited by the continuous wave pumping laser. This balance between particle loss and pump creates a microscopically populated state (the polariton BEC) with temporal coherences orders of magnitude longer than the bare photon coherence of about 10 ps. The non-equilibrium nature enables the formation and coexistence of polariton BECs in different confined states of the trap, as illustrated in Fig. 1(d). The relative particle population of these states is dictated by the balance between stimulated scattering and particle losses, rather than by thermodynamical arguments.

- To address the referee’s question, we added the following text to page 6:

The simultaneous formation of condensates both in the GS and ES is a consequence of the non-equilibrium nature of polariton BECs. Unlike a conventional BEC in thermodynamical equilibrium, polariton BECs result from the dynamic balance between particle losses (e.g., due to the photon escape from the MC) and replenishment by stimulated scattering from states excited by the pumping laser. This non-equilibrium process enables the formation and coexistence of BECs in different confined states of the trap with particle densities dictated by the dynamics of the stimulated scattering and particle loss mechanisms.

- Regarding the period of the spectral oscillations in Fig. 1(f): please note that this period is indeed the SAW frequency T_M , and not 1 ns (note that the time unit in the horizontal axis is T_M). Fig. 1(f) is thus fully consistent with the modulation model. To make that clear, we added the following sentence to the caption of Fig. 1:

”The time scale in panel (f) is in units of the SAW period, $T_M = 1/f_M$.”

Point 2.2:

”The 2D representation chosen in Fig3a is hard to decipher, and it is not clear how the spectra of the other panels are derived from such maps. Were the data of Fig2 acquired in the same way? If so, the same choice of representation should be kept in both figures, in order to make it clear that only the SAW amplitude is being varied. From this figure, it appears that the integrated emission from the BEC excited state is higher than that of the ground state, contrary to what is shown in Fig1. Is this an artifact of the measurement technique, or is their more to it?”

Response:

All high resolution spectra acquired with the etalon were derived from maps similar to the one in Fig. 3(a) of the previous version of the manuscript. We agree with the referee about the importance of having the same representation for all data in order to avoid misunderstanding. In the revised version of the manuscript, we introduced the following modifications:

- In order to clarify the measurement procedure, we introduced a new section in the supplement (Sec. SM1, ”High-resolution spectroscopy on polariton BECs”) describing in more details the experimental procedure for the acquisition of PL data with sub-GHz resolution. The PL map of Fig. 3(a) of the previous version was transferred to this section.
- The text (and caption) associated with Fig. 3 was correspondingly modified to account for the changes.

In agreement with the referee’s assertion, the PL amplitude from the excited state (ES) in Fig. 3 is indeed higher than for the ground state (GS), as show in Fig. SM1(b) of the supplement. This is not an artifact: it results from the non-equilibrium nature of the confined BECs, as addressed in Point 1 above. We chose, however, to display the PL amplitudes normalized to their maximum values.

Point 2.3:

” In Fig.3c, the data and simulation are quite off for the negative energy shift regime. It is argued in the main text that this discrepancy could be attributed to strain-induced admixture between the BEC excited state and other confined levels. First, it is not clear from Fig1 that such higher order confined levels still exist in the trap at the pump fluence used in the experiment due to the mean-field energy shift of the polariton BEC. Second, the authors should justify why such an effect would only affect the negative energy shift part of the spectrum. ”

Response:

- We first clarify the question about the depth of trap potential: the present studies used a trap with a lateral confinement potential of 5 meV, which is almost 20 times larger than the maximum modulation amplitude (of approx. 0.35 meV) employed in the experiments. This potential holds several confined states, as illustrated in Fig. 1(c). The modifications introduced in the text to address this issue are summarized in Point 3.3 of Referee 3.
- The experiments were carried out on a polariton trap placed within an acoustic resonator consisting of two interdigital transducers. The resonator supports several longitudinal modes: in the present investigations, care was taken to tune the SAW frequency to a mode with an antinode aligned with the trap center. Under this condition, the SAW strain field has the least impact on mode coupling and the modulation becomes approximately symmetric for both the ground and excited states. Slight shifts of the anti-node with respect to the trap center increase the asymmetry, in particular for the more extended excited states. These shifts can lead to differences in the positive and negative energy shifts.

In order to call attention to the asymmetry and explain its origin, we have introduced the following modification in the text of page 13:

The model reproduces remarkably well almost all spectral features of the GS, including an excellent fit of the acceleration beats at the extrema of the energy modulation of the GS [cf. right inset of Fig. 3(a)]. The same applies to the blue-shifted acceleration beats for the ES as illustrated by the right inset of Fig. 3(b). In particular, we found that the amplitudes of the calculated acceleration beats are very sensitive to BEC temporal coherence. The excellent agreement between model and the experiments thus also proves that the acoustic pumping can induce very large energy modulation amplitudes (e.g., $\chi^{(s)} > 200$) without introducing decoherence.

While the model predicts symmetric energy shifts for positive and negative sidebands, the negative energy shift for the ES is larger than the positive one [cf. Fig. 3(b)]. This asymmetry arises from quadratic terms in the dependence of the polariton energy shifts on the modulation amplitude, which are not taken into account by the linear dependence expressed by Eq. (2) and are more pronounced for the ES. The quadratic terms are attributed to a spatially asymmetric distribution of the strain field in the trap location. If the polariton trap is exactly at an anti-node of the hydrostatic SAW strain field, this field has the least impact on the trap symmetry, yielding almost equal positive and negative energy shifts. Slight shifts of the trap center relative to the anti-node can, however, induce large quadratic contributions to the energy modulation (cf. Secs. SM2 and SM5). These contributions depend on the symmetry and extension of the BEC wave function and lead to strain-induced admixtures of confined levels [22], which are normally more pronounced for the ES than for the GS. We show in Sec. SM5 that shifts of the anti-node with respect to the trap center of less than one μm can already introduce the asymmetry observed for the ES in Fig. 3(b) while maintaining the GS energy shifts almost symmetric.

- The effects of shifts in the position of the trap relative to the SAW anti-nodes, which yields the main contribution to the asymmetry, is now quantitatively described in a new section of the supplement (Sec. SM5).
- We also addressed a second mechanism for the asymmetry, which is related to the modulation of the confinement potential depth, by modifying the following text of Sec. SM2.

We determined the modulation coefficients $\hbar\Omega_M\chi^{(s)}$ and $\hbar\Omega_M\chi'^{(s)}$ by solving Eq. (SM1) using the measured detuning and Rabi coupling for the trap states. Figure SM2(a) displays the variation of the lower (LP) and upper polariton (UP) as a function of the static detuning $\delta_{CX,0}$. The lower panel of the figure shows the dependence of $\chi^{(s)}$ and the ratio $\chi'^{(GS)}/\chi^{(GS)}$ on detuning. The green dot marks the detuning for the intracavity trap used in the experimental studies and the small arrows around it approximate the amplitude of the strain-induced energy modulation for the conditions in Fig. 3 of the main text (corresponding to $\chi^{(GS)} = 220$). The quadratic contribution in Eq. (SM2) for these small modulation amplitudes is much smaller than the linear one. This result justifies neglecting this term in the analysis of the sidebands presented in the main text. We note, however, that the quadratic term is also expected to induce an overall small red-shift of the sideband spectrum by $(1/2)\hbar\Omega_M\chi'^{(s)}$ estimated to be equal to 5 and 4 GHz for the GS and ES, respectively, for $\chi^{(GS)} = 220$. The

small magnitude and the approximately equal values for the GS and ES makes it difficult to discriminate these shifts from, e.g., thermal drifts due to the acoustic excitation. For these reasons, the quadratic contribution can also not account for the asymmetric shape of the sideband spectrum for the ES in Fig. 3(b) of the main text.

Point 2.4:

"The discussion of the theoretical model for the interaction between SAWs and the polariton BEC is too short and technical for a reader who is not expert in the field. In particular, the simplifications made regarding the strain-induced radiation pressure effects and the hydrostatic component should be given more justifications in order for the reader to understand the physical assumption of the model used in the manuscript, without having to dive into the SAW literature."

Response:

We added the following text on the page 6:

In agreement with previous studies [16], the energy modulation amplitude is mainly determined by the strain-induced changes of the excitonic energies via the deformation potential (DP) mechanism (i.e., the changes of the excitonic energies induced by a strain field, see, for details, Sec. SM2).

The following paragraph was included in the beginning of Sec. SM2:

In a simplified way, the coupling of acoustic strain to a polariton state can be described via strain-dependent detuning between bare photon (C) and exciton (X) resonances: $\Delta E(\eta) = C(\eta) - X(\eta)$, where η is the strain for a particular SAW phase. The individual energies can be written as $X(\eta) = X_0 + a_h \times \eta$ and $C(\eta) = C_0 + \beta \times \eta$, where $a_h = -9$ eV is the GaAs hydrostatic deformation potential, X_0 and C_0 are the unstrained exciton and photon energies. β is the shift of the cavity resonance per unit of strain due to modifications of the spacer thickness and refractive index, which depends on the strain distribution and cavity geometry. It turns out that in similar microcavities, the ratio of the relative changes of exciton and photon energies under strain $\Delta X(\eta)/\Delta C(\eta)$ is in the range from 3 to 20 [25, 26]. Therefore, the effect of strain on the photon energy plays a secondary role.

Point 2.5:

"Lines 198 and 205, "Eq.(7)" should be replaced by "Eq.(6)"

Response:

We have checked equation labels and corrected errors.

Reviewer 3 remarks

"This paper reports the observation of interference effects in the optical emission spectrum induced by the rapid energy modulation applied to polariton BEC states. The modulation is induced by the periodic acoustic strain through SAW excitation and the phase delay/advance caused by the frequency acceleration between two time-durations sandwiching the highest strain point causes the destructive and constructive interference, which leads to the periodic PL intensity modulation near the spectral band edges. The physics behind is clear and interesting, and it is nice to see the effect of temporal coherence even though the polariton decoherence time is shorter than the period of strain modulation, so that I believe it can give a new universal insight into the hybrid systems consisting of highly frequency-mismatched systems. However, there are points to be clarified and I recommend the authors to improve the manuscript before recommending the publication."

Point 3.1:

”The role of phase decoherence is not clear. If I correctly understand, the theoretical framework is first based on the perfectly coherent system and the effect of decoherence is then introduced simply by using the Gaussian smoothing. I believe that this is very rough discussion because the suppression of interference is not caused by something like inhomogeneous broadening but by the phase decoherence. In fact, I guess that the observed interference is suppressed also when the phase decoherence time is much larger than the SAW period. This is because the incommensurate relation between the SAW and PL frequencies causes different phase shift between the two time-durations with a similar emission energy but separated longer than one period. (Of course, side bands start to appear in this situation, but the averaging should show the similar trend with suppressed interference.) Therefore, the interference pattern is observed only within the case when the decoherence time is shorter than the period but larger than the time required to induce the interference. Then, I wonder if simply using the Gaussian smoothing is not suitable for describing this system. I recommend the authors to describe the model based on the phase decoherence and clarify the condition to observe the effects. (They may be performed finite-time integration in Fourier transformation, but it is not explicitly mentioned.)”

Response:

The situation described by the referee, namely, modulation periods far shorter than the BEC coherence time was experimentally accessed using much higher acoustic frequencies in a recent paper from our group (ref. [14]). The modulation induces several non-adiabatic narrow subbands without introducing noticeable decoherence. The mechanism mentioned by the referee (i.e., incommensurability between the modulation frequency and BEC energy) apparently does not play a role. We attribute this behavior to the fact that the highly nonlinear confined BEC can easily adjust its energy (via small changes in the particle density) to always remain commensurate with the modulation frequency.

In the present case, the SAW period exceeds the BEC coherence time. The theoretical framework expressed by equations 2-8 was indeed derived for a perfectly coherent system: the effect of decoherence was then introduced simply by using the Gaussian smoothing. The smoothing was included in the theoretical curves of Figs. 2 and 3 to account for the linewidth of the BEC line in the absence of acoustic excitation, which, as illustrated in these figures, is well-described by a Gaussian curve. This important results emphasized in the text is that one reproduces very well the sidebands spectra with accelerations beats using exactly the same linewidth as in the absence of the acoustic modulation, thus implying that the acoustic excitation introduces negligible decoherence.

To clarify this points, we he following modification on page 13:

The blue curves in Figs. 2 and 3 were determined by convoluting the sideband spectra obtained from to Eq. (6) with a Gaussian lineshape with a FWHM exactly equal to the one measured for the BEC in the absence of acoustic excitation in order to account for the finite temporal coherence of the BEC. Most of the spectral features are determined by the $\chi^{(s)}$. The ΔM merely introduces a reduction in the total emission intensity as well as an asymmetry in the intensity of the positive and negative sidebands (note that the term related to ΔM in Eq. (6) does not change sign when under time reversal, while the others do). These results contrast with reports for electrically modulated vertical cavity lasers, where the sidebands are primarily induced by the ΔM factor [21].

Point 3.2:

”The onset of modulation amplitude to observe this phenomenon gives a quantitative knowledge on the decoherence time. If the author can include this discussion, it would be more interesting and useful. Probably, the model for discussing the phase decoherence is required. ”

Response:

This question has been addressed within Point 1 above.

Point 3.3:

”The SAW modulation for the energy shift of 100GHz is large and comparable to the polariton trapping potential, possibly inducing the nonlinear energy shift. Does this cause additional effects? ”

Response:

The trap confinement potential, which is now displayed by the solid curve superimposed on Fig. 1(c), has a depth $\Delta E_{Pot} \approx 5$ meV (or 1200 GHz).The confinement potential (E_{Pot}) can be directly determined

from the trap dimensions and the energies of the confined levels of Fig. 1(c) relative to the trap barrier at 1534 meV. This potential, which is now displayed by the solid curve superimposed on Fig. 1(c), has a depth $\Delta E_{Pot} \approx 5$ meV (or 1200 GHz). The potential depth is thus at least an order of magnitude larger than the SAW-induced amplitude energy shifts in Fig. 1(f) and two orders of magnitude larger than the energy modulation in the BEC regime of Figs. 2 and 3.

The following changes were included in the manuscript to address this point:

- The potential profile determined for the trap is now superimposed on the PL map of Fig. 1(c) together with the lower-lying confined wave functions. The following text was added to the caption: **The solid and dashed curves display the trap confinement potential and squared wavefunctions (for the lower confined levels), respectively, determined as described in Sec. SM5.**
- The new section **SM5** now describes the procedure employed for the determination of the confinement potential and confined wave functions.
- We added the following text addressing the above point on the page 8:
We note that this modulation amplitude is more than an order of magnitude smaller than the depth of the confinement potential of 5 meV [cf. Fig. 1(c)].

Point 3.4:

”The role of BEC is unclear. It is required to have enough long decoherence time? ”

Response:

The polariton linewidth reduces from approximately 0.2 meV [cf. Fig. 1(c)-(f)] to less than 4 μ eV in the condensation regime. The role of the condensate is thus to increase coherence while maintaining strong energy sensitivity to the strain. To clarify the role of the polariton BEC, we added the following text to the introduction on the page 4:

At low particle densities, the polariton coherence is determined by the exciton decoherence and the cavity photon lifetime, leading to spectral linewidths of typically a few hundreds of μ eV. At high densities, polaritons undergo a transition to a highly coherent *non-equilibrium* Bose-Einstein condensate (BEC) with linewidths down to a few μ eV and temporal coherence reaching the ns-range. In the BEC regime, polaritons retain an enhanced coupling to strain fields via their excitonic component, which enables efficient energy modulation by acoustic waves.

Point 3.5:

*”There are many typos. The author should much more carefully prepare the manuscript. For examples,
- Many index (s) depicting the polariton mode is replaced by (p): Line 158, 160, 345, 346, 3rd line above 354, 354, etc. Both can be used by should be unified.
- Duplicate words: “again” in line 128, “using the” in 213, “potential” in 336.
- Numbering of Figures: Use ones of 3a, 3b, 3c or 3(a), 3(b), 3(c) for example in page 7. - Do you need two pi in line 160?
- In many parts, energy and frequency is confusingly used. Insert or delete hbar correctly. For example, Fig 2(c), line 160, 201, SM1, SM2, below 353, 354,
- In line 196, m is missing.
- Line 198, Eq.(7) should be (5) or (6)?
- small pi in line 372.
- Fig. SM2 caption: (a) 0.2 \rightarrow 0 and (b) the value of DeltaM_PL should be shown (maybe 0.2?).
- Schrodinger equation SM5 does not include the wavefunction in R.H.S.”*

Response:

We thank the Referee for detecting these deficiencies. In the revised manuscript we did our best to correct typos and other formatting deficiencies.

Additional remarks

In addition to the modifications listed above, the following changes were also included in the revised version:

- Typos were corrected through the text

- Labels in several figures were corrected
- Some of the equations were re-written to improve readability
- New figures (Figs. 4 and 5 of the main text as well as Figs. SM4, SM6, and SM7 of the supplement) were introduced/modified.
- Equations Eq. 2 and Eq. SM2 as well as the text around them was rewritten to state the modulation parameters in terms of the exciton Hopfield coefficient, which has an established physical meaning.

-
- [1] S. H. Autler and C. H. Townes, Stark effect in rapidly varying fields, *Phys. Rev.* **100**, 703 (1955).
- [2] M. Aspelmeyer, T. J. Kippenberg, and F. Marquardt, Cavity optomechanics, *Rev. Mod. Phys.* **86**, 1391 (2014).
- [3] V. Torres-Company and A. M. Weiner, Optical frequency comb technology for ultra-broadband radio-frequency photonics, *Laser & Photonics Reviews* **8**, 368 (2013).
- [4] B. Zaks, R. B. Liu, and M. S. Sherwin, Experimental observation of electron-hole recollisions, *Nature* **483**, 580 (2012).
- [5] D. A. Golter, T. Oo, M. Amezcua, K. A. Stewart, and H. Wang, Optomechanical quantum control of a nitrogen-vacancy center in diamond, *Phys. Rev. Lett.* **116**, 143602 (2016).
- [6] B. Pigeau, S. Rohr, L. M. de Lépinay, A. Gloppe, V. Jacques, and O. Arcizet, Observation of a phononic mollow triplet in a multimode hybrid spin-nanomechanical system, *Nat. Commun.* **6**, 10.1038/ncomms9603 (2015).
- [7] W. J. M. Naber, T. Fujisawa, H. W. Liu, and W. G. van der Wiel, Surface-acoustic-wave-induced transport in a double quantum dot, *Phys. Rev. Lett.* **96**, 136807 (2006).
- [8] M. Metcalfe, S. M. Carr, A. Muller, G. S. Solomon, and J. Lawall, Resolved sideband emission of InAs/GaAs quantum dots strained by surface acoustic waves, *Phys. Rev. Lett.* **105**, 037401 (2010).
- [9] B. Villa, A. J. Bennett, D. J. P. Ellis, J. P. Lee, J. Skiba-Szymanska, T. A. Mitchell, J. P. Griffiths, I. Farrer, D. A. Ritchie, C. J. B. Ford, and A. J. Shields, Surface acoustic wave modulation of a coherently driven quantum dot in a pillar microcavity, *Appl. Phys. Lett.* **111**, 011103 (2017), <https://doi.org/10.1063/1.4990966>.
- [10] D. Wigger, M. Weiß, M. Lienhart, K. Müller, J. J. Finley, T. Kuhn, H. J. Krenner, and P. Machnikowski, Resonance-fluorescence spectral dynamics of an acoustically modulated quantum dot, *Phys. Rev. Research* **3**, 033197 (2021).
- [11] R. A. DeCrescent, Z. Wang, P. Imany, R. C. Boutelle, C. A. McDonald, T. Autry, J. D. Teufel, S. W. Nam, R. P. Mirin, and K. L. Silverman, Large single-phonon optomechanical coupling between quantum dots and tightly confined surface acoustic waves in the quantum regime, *Phys. Rev. Appl.* **18**, 034067 (2022).
- [12] T. Berstermann, A. V. Scherbakov, A. V. Akimov, D. R. Yakovlev, N. A. Gippius, B. A. Glavin, I. Sagnes, J. Bloch, and M. Bayer, Terahertz polariton sidebands generated by ultrafast strain pulses in an optical semiconductor microcavity, *Phys. Rev. B* **80**, 075301 (2009).
- [13] D. L. Chafatinos, A. S. Kuznetsov, S. Anguiano, A. E. Bruchhausen, A. A. Reynoso, K. Biermann, P. V. Santos, and A. Fainstein, Polariton-driven phonon laser, *Nat. Commun.* **11**, 4552 (2020).
- [14] A. S. Kuznetsov, K. Biermann, A. A. Reynoso, A. Fainstein, and P. V. Santos, Microcavity phonoritons: a coherent optical-to-microwave interface, *Nat. Commun.* **14**, 5470 (2023), arXiv:2210.14331v1.
- [15] A. S. Kuznetsov, P. L. J. Helgers, K. Biermann, and P. V. Santos, Quantum confinement of exciton-polaritons in a structured (Al,Ga)As microcavity, *Phys. Rev. B* **97**, 195309 (2018).
- [16] A. S. Kuznetsov, K. Biermann, and P. V. Santos, Dynamic acousto-optical control of confined polariton condensates: From single traps to coupled lattices, *Phys. Rev. Research* **1**, 023030 (2019).
- [17] A. S. Kuznetsov, P. L. J. Helgers, K. Biermann, and P. V. Santos, Quantum confinement of exciton-polaritons in structured (Al,Ga)As microcavity, *Phys. Rev. B* **97**, 195309 (2018).
- [18] T. Sogawa, P. V. Santos, S. K. Zhang, S. Eshlaghi, A. D. Wieck, and K. H. Ploog, Dynamic band-structure modulation of quantum wells by surface acoustic waves, *Phys. Rev. B* **63**, 121307 (2001).
- [19] A. S. Kuznetsov, D. H. O. Machado, K. Biermann, and P. V. Santos, Electrically driven microcavity exciton-polariton optomechanics at 20 GHz, *Phys. Rev. X* **11**, 021020 (2021), <https://arxiv.org/abs/2003.01051>.
- [20] P. Yu and M. Cardona, *Fundamentals of Semiconductors: Physics and Materials Properties* (Springer, Heidelberg, 1995).
- [21] X. Zhu and D. T. Cassidy, Modulation spectroscopy with a semiconductor diode laser by injection-current modulation, *J. Opt. Soc. Am. B* **14**, 1945 (1997).
- [22] A. S. Kuznetsov, G. Dagvadorj, K. Biermann, M. H. Szymanska, and P. V. Santos, Dynamically tuned arrays of polariton parametric oscillators, *Optica* **7**, 1673 (2020), <http://arxiv.org/abs/2003.01386>.
- [23] O. E. Daïf, A. Baas, T. Guillet, J.-P. Brantut, R. I. Kaitouni, J. L. Staehli, F. Morier-Genoud, and B. Deveaud, Polariton quantum boxes in semiconductor microcavities, *Appl. Phys. Lett.* **88**, 061105 (2006).
- [24] R. I. Kaitouni, O. El Daïf, A. Baas, M. Richard, T. Paraiso, P. Lugan, T. Guillet, F. Morier-Genoud, J. D. Ganière, J. L. Staehli, V. Savona, and B. Deveaud, Engineering the spatial confinement of exciton polaritons in semiconductors, *Phys. Rev. B* **74**, 155311 (2006).
- [25] E. A. Cerda-Méndez, D. N. Krizhanovskii, K. Biermann, R. Hey, P. V. Santos, and M. Skolnick, Effects of the piezoelectric field in the modulation of exciton-polaritons by surface acoustic waves, *Superlattices Microstruct.* **49**, 233 (2011).
- [26] A. V. Scherbakov, T. Berstermann, A. V. Akimov, D. R. Yakovlev, G. Beaudoin, D. Bajoni, I. Sagnes, J. Bloch, and M. Bayer, Ultrafast control of light emission from a quantum-well semiconductor microcavity using picosecond strain pulses, *Phys. Rev. B* **78**, 241302 (2008).
- [27] D. C. Valovcin, H. B. Banks, S. Mack, A. C. Gossard, K. West, L. Pfeiffer, and M. S. Sherwin, Optical frequency combs from high-order sideband generation, *Optics Express* **26**, 29807 (2018).

REVIEWERS' COMMENTS

Reviewer #1 (Remarks to the Author):

In the majorly revised resubmission of the manuscript the authors have addressed most points raised in the referees reports providing additional data where needed. They have also clarified the differences and novelty of their observations in relation to the previous reports. I therefore recommend the paper for publication in Nature Com journal.

Reviewer #3 (Remarks to the Author):

The authors made appropriate revision in response to my suggestions. I recommend the publication.